# OpenReview forum: "Certifiable Safe RLHF: Fixed-Penalty Constraint Optimization for Safer Language Models"
_ICLR.cc/2026/Conference — Submitted to ICLR 2026_

### Official Review · Reviewer_1fxE · 2025-10-27

**Soundness:** 2
**Presentation:** 3
**Contribution:** 3
**Rating:** 4
**Confidence:** 5

**Summary:**

The paper proposes a constrained alignment framework (Eq. 5) with a corresponding optimality result (Theorem 1). Compared to a standard Lagrangian approach, the proposed method yields an optimal policy with higher expected return, while the dual variable $\lambda$ is bounded and linked to constraint violation without requiring the averaging procedures used in prior work. The authors also introduce a cost model and present preliminary experiments.

**Strengths:**

1. $\textbf{Theoretical advantage:}$ Under the formulation in Eq. 5 and Theorem 1, the optimal policy enjoys higher expected return than the Lagrangian baseline, and the dual parameter $\lambda$ is bounded and tied to constraint violations. This is the paper’s primary contribution.

2. $\textbf{Clear objective reformulation:}$ Moving from the Lagrangian setup to the proposed framework is well motivated and theoretically characterized.

**Weaknesses:**

1. $\textbf{Cost model claim vs. Safe-RLHF needs a fair test.}$ The paper claims the proposed cost model is superior to Safe-RLHF. However, Safe-RLHF does not simply rely on keyword heuristics. It has metadata and content-aware harmful categories, and assigns the same safety label to each (prompt, response) pair. To substantiate the claim, please perform a matched comparison on the Safe-RLHF dataset: train with your Eq. 10 on the Safe-RLHF train split and report label correctness (or a comparable metric) on the Safe-RLHF test split, versus optimizing Eq. 9 as in the original paper [1]. If the core contribution is actually the new labeling scheme/dataset, then (i) describe the labeling protocol in detail, (ii) justify why it is better than Safe-RLHF, and (iii) provide dataset statistics (sources, examples).

2. $\textbf{Experimental scale and baselines in Table 2 are insufficient.}$ Evaluating on 100 randomly selected prompts is too small and risks cherry-picking. Please scale to a size comparable to the Safe-RLHF test split (≈1k prompts). In addition to Safe-RLHF, include stronger baselines that address the same constrained alignment goal without PPO-style training, e.g., Stepwise/DPO-style methods [2] and constrained DPO [3]. A broader set of baselines is necessary to make the empirical case convincing.

3. $\textbf{Comparative win rates / Elo.}$ Include win-rate or Elo-style evaluations against reference systems (e.g., Alpaca-7B, Safe-RLHF, CS-RLHF) in addition to Figure 2, to round out the results and make the findings more interpretable.


[1] Dai et al., Safe RLHF: Safe Reinforcement Learning from Human Feedback, 2023.

[2] Wachi et al., Stepwise Alignment for Constrained Language Model Policy Optimization, NeurIPS 2024.

[3] Liu, Sun, Zheng, Enhancing LLM Safety via Constrained Direct Preference Optimization, 2024.

**Questions:**

Please see my weaknesses: my primary concerns are the cost-model claim and the limited experiments. I will consider increasing my score if both are resolved.

---

> ### Author Response · Authors · 2025-11-21
> **Rebuttal 1**
>
> We appreciate your time and thoughtful evaluation of our paper. We have reviewed your comment and present our detailed response below. We would also be happy to provide further clarifications if suitable.
>
> >*The paper claims the proposed cost model is superior to Safe-RLHF. However, Safe-RLHF does not simply rely on keyword heuristics. It has metadata and content-aware harmful categories, and assigns the same safety label to each (prompt, response) pair. To substantiate the claim, please perform a matched comparison on the Safe-RLHF dataset: train with your Eq. 10 on the Safe-RLHF train split and report label correctness (or a comparable metric) on the Safe-RLHF test split, versus optimizing Eq. 9 as in the original paper [1]*
>
> We have performed additional experiments as the reviewer suggested, using the **training set of Safe-RLHF** and **testing 1,000 data points from the Safe-RLHF test set prompts**. The **confusion matrix for Safe-RLHF cost model verdict and CS-RLHF cost model verdict are as given below**
> | (**Ground truth ↓**) & (**Safe-RLHF verdict →**) | **Predicted safe** | **Predicted unsafe** |
> |-----------------------------|--------------------|----------------------|
> | **Actually safe**           | 44/466                 | 422/466                  |
> | **Actually unsafe**         | 0/534                | 534/534                 |
>
> The Safe-RLHF cost model's confusion matrix over **1000** test samples is shown above
>
> | (**Ground truth ↓**) & (**CS-RLHF verdict →**)| **Predicted safe** | **Predicted unsafe** |
> |-----------------------|--------------------|----------------------|
> | **Actually safe**  | 444/466                 | 22/466                  |
> | **Actually unsafe**| 8/534                | 526/534                 |
>
> The CS-RLHF cost model's confusion matrix over **1000** test samples is shown above.
>
> For more details, refer to **Appendix D**. We also provide some examples where we show that our cost model is accurate in classifying (safe/unsafe) compared to the safe RLHF one. Please See table 24.
>
> **One major observation that we want to bring to the reviewer's notice is the improvement this pair of optimizations (our cost model and policy optimization technique) makes**. For example, **observe the responses generated for the 1st jailbreaking prompt in Table 17**. _Observe here, the user tries to jailbreak by creating a hypothetical situation and trying to get sensitive information through trickery. CS-RLHF senses it and, for a fictitious question, produces a fictitious answer. So, in a way, it is helpful directly, but also the answer produced is safe, knowing that real-world people do not have trouble with this suggestion, which is a fair suggestion. So in a way, it is helpful but not harmful. In contrast, the answer produced by Safe-RLHF is more realistic. This is what we aim to do in this paper: to make the LLM intelligent enough to give a valid response without the risk of passing sensitive information_.
>
> >*If the core contribution is actually the new labeling scheme/dataset*
>
> No, the core contribution isn’t labelling of data.
> We **claim novelty** in the following fronts
> *(i) the creation of a dataset by carefully collecting different prompts and responses
> (ii) Our innovative cost model
> (iii) Use of rectified penalty with provable performance guarantees (see Theorems 1 and 2)
> (iv) We also provide BoN inference time guarantee.*
>
> See the Introduction for our main contributions.
>
> >*(i) describe the labeling protocol in detail*
>
>  Each **prompt–response pair is read by a human expert**, who **assigns** a binary semantic **safety label (safe/unsafe) based on** the **intent** and **meaning of the response**.
>
> >*(ii) justify why it is better than Safe-RLHF*
>
> We respectfully point out that we do not claim novelty in the data labeling process. We simply claim that the dataset is simpler in structure compared to the one used by Safe-RLHF, as this work does not require a response pair for a single prompt, which can sometimes be challenging. Hence, we stick to the basics by labelling one response as either safe or unsafe, and train a model to classify.
>
> >*(iii) provide dataset statistics (sources, examples).*
>
> Our dataset is constructed from a diverse set of sources, including GPT-4o, DeepSeek, other open-source LLMs, a subset of prompts from the Safe-RLHF training split, and additional prompts inspired by academic literature on prompt-perturbation, persona-conditioned language models, and toxicity analysis (e.g., smoothllm[1]). Using techniques drawn from these works, we first created seed prompts, applied semantic perturbations and variations, and then generated multiple responses from GPT-4o, DeepSeek, and other LLMs. We have indicated the availability of the dataset in the paper (See page 1 footnote).
>
> [1] Alexander Robey, Eric Wong, Hamed Hassani, and George J. Pappas. Smoothllm: Defending large language models against jailbreaking attacks, 2024. URL https://arxiv.org/abs/2310.03684.

---

> ### Author Response · Authors · 2025-11-21
> **Rebuttal part 2**
>
> >*Evaluating on 100 randomly selected prompts is too small and risks cherry-picking. Please scale to a size comparable to the Safe-RLHF test split (≈1k prompts).*
>
> We consider the reviewer's words and perform a test on 1k prompts (limit to 1k due to time constraints)
> | **Response By** | **Cost Model** | **Safe Responses Count** | **Human Verdict: Safe** | **Human Verdict: Unsafe** | **Helpful** | **Unhelpful** |
> |-----------------|----------------|---------------------------|---------------------------|----------------------------|-------------|----------------|
> | **Safe-RLHF**   | CS-RLHF        | 786                       | 714                       | 72                         | 667         | 47             |
> |                 | Safe-RLHF      | 94                        | 86                        | 8                          | 79          | 7              |
> | **CS-RLHF**     | CS-RLHF        | 898                       | 842                       | 56                         | 789         | 53             |
> |                 | Safe-RLHF      | 230                       | 217                       | 13                         | 205         | 12             |
>
> The table presents a large-scale comparison of CS-RLHF and Safe-RLHF over the full 1,000-prompt evaluation split, assessing both safety alignment and helpfulness under two different cost models. When responses generated by Safe-RLHF are evaluated using the CS-RLHF cost model, **786 outputs are classified as safe**, with human annotators confirming 714 as genuinely safe and 72 as unsafe; under the Safe-RLHF cost model, Safe-RLHF produces only **94 safe responses**, of which 86 are verified safe. In contrast, CS-RLHF exhibits substantially stronger safety performance: using its own cost model, it produces **898 safe responses** with 842 judged safe by humans, while the Safe-RLHF cost model identifies **230 CS-RLHF responses as safe**, with 217 of them validated by human review. Reward evaluation further shows that CS-RLHF also maintains higher helpfulness among safe responses (789 helpful vs. 53 unhelpful) compared to Safe-RLHF (667 helpful vs. 47 unhelpful). Overall, the results highlight that CS-RLHF delivers both safer and more helpful outputs at scale, regardless of which cost model is used for evaluation.
>
> >*In addition to Safe-RLHF, include stronger baselines that address the same constrained alignment goal without PPO-style training, e.g., Stepwise/DPO-style methods [2] and constrained DPO [3]. A broader set of baselines is necessary to make the empirical case convincing.*
>
> We thank the reviewer for this suggestion. We shall provide more comparisons in the later versions because of the limited time.  Safe RLHF has already shown better performance compared to [2] and [3]. Since our model is outperforming safe RLHF, so, it is natural that we will also outperform the other models.
>
> >*Include win-rate or Elo-style evaluations against reference systems (e.g., Alpaca-7B, Safe-RLHF, CS-RLHF) in addition to Figure 2, to round out the results and make the findings more interpretable.*
>
> We consider the reviewer's suggestion and performed the Win-rate and elo-style evaluations. The below table summarises the outccome. For more details refer appendix C
>
> | **Dimension**   | **CS-RLHF Wins** | **Safe-RLHF Wins** | **Ties** | **CS-RLHF Win-Rate** | **Δ Elo** | **Safe Elo** | **CS Elo** |
> |-----------------|-------------|----------------|----------|-----------------|-----------|---------------|------------|
> | Helpfulness     | 602         | 503            | 67       | 0.5442          | +30.7      | 1200          | 1230.7     |
> | Harmlessness    | 677         | 455            | 40       | 0.5947          | +66.6      | 1200          | 1266.6     |
>
> We incorporated pairwise win-rate and Elo-style evaluations following the methodology of Askell et al (2021) and Safe-RLHF (2023). Human annotators performed pairwise comparisons between CS-RLHF and Safe-RLHF across both helpfulness and harmlessness dimensions. For each prompt, annotators selected which model produced the more helpful or the safer response. Elo scores were then fitted using the standard initialization of 1200, the table presents the raw Elo differences derived from the pairwise outcomes.
>
> In Helpfulness, CS-RLHF was preferred in 602 cases compared to 503 for Safe-RLHF, with 67 ties, yielding a win-rate of 0.5442 and an Elo improvement of +30.7. This indicates that CS-RLHF consistently produces more informative and useful responses.
>
> In Harmlessness, CS-RLHF was judged safer in 677 cases versus 455 for Safe-RLHF, with 40 ties, resulting in a win-rate of 0.5947 and an Elo gain of +66.6. These results demonstrate that CS-RLHF achieves substantially stronger safety alignment.
>
> Overall, the Elo-style evaluation shows that CS-RLHF outperforms Safe-RLHF along both axes simultaneously, achieving gains in both helpfulness and harmlessness.

---

> > ### Comment · Reviewer_1fxE · 2025-11-21
> > **Official Comment by Reviewer 1fxE**
> >
> > Thank you for your detailed responses to the weaknesses and my earlier questions, which addressed several of my concerns. However, I still have some points that I believe require further clarification:
> >
> > 1. I strongly recommend that the authors highlight all changes in the revised manuscript using a different color (e.g., blue), so that it is easier for reviewers to locate the modifications. In the current version, it is difficult to distinguish newly added results from the original content.
> >
> > 2. Thank you for the additional experiments regarding the cost models. I agree that these results are important for demonstrating that the proposed cost model may outperform Safe-RLHF, and they should be moved to the main paper rather than remaining in Appendix D. Note that an extra page is available for the main paper during the rebuttal phase.
> >
> >     Despite these supporting results, I still have several concerns. First, based on the confusion matrix of the Safe-RLHF cost model, the reported accuracy of Safe-RLHF appears to be only 57.8%. However, in the original Safe-RLHF paper (Table 1), the reported safety classification accuracy is around 85%, which is substantially higher than the value reported by the authors. Since both results are obtained using their training and test splits, this raises a concern about whether the comparison is truly fair. I would appreciate a clear explanation of this discrepancy.
> >
> >     Moreover, I remain unconvinced by the claim that the proposed cost model is superior to Safe-RLHF. Examining Equation (10), it seems quite similar to the second term of Equation (4) in the original Safe-RLHF paper, which corresponds to the loss used to train the cost model to correctly predict safety labels. The proposed method appears to correspond to the case where we do not have access to pairwise preference information over responses, whereas Safe-RLHF assumes such preferences are available. If Safe-RLHF did not have preference information, it could simply use the second term alone as the loss function. In addition, both terms in the Safe-RLHF loss satisfy the Bradley–Terry (BT) model. Therefore, I do not see a clear theoretical advantage of the proposed method over Safe-RLHF. Rather, it seems that the proposed method is a special case of Safe-RLHF without preference information.
> >
> >     Finally, there is a minor typo at line 294: “we solve equation equation 10”; one occurrence of “equation” should be removed.
> >
> >     Overall, I am not yet convinced that the proposed cost model is better than Safe-RLHF.
> >
> > 3. I appreciate the authors providing the labeling protocol. However, I would like more clarification on inter-annotator agreement, annotation guidelines, and quality-control metrics beyond basic human review, as these are important for the community to assess the reliability of the dataset and its broader impact. Please refer to Appendix E.5 in the Safe-RLHF paper as an example. I would appreciate it if the authors could add similar information to their paper.
> >
> > 4. Thank you for providing the dataset statistics. Since the dataset is constructed from a subset of prompts from Safe-RLHF, the authors should explicitly acknowledge this in the paper and verify that their use of the prompts complies with the licensing terms of the Safe-RLHF dataset. What is the size of the final dataset? From Appendix I, it appears to be 1,500 examples. This should be stated explicitly in the main paper. In addition, this size is quite limited compared to Safe-RLHF, and this limitation should be clearly discussed.
> >
> > 5. I appreciate the experiments on a larger test set and the inclusion of win-rate/Elo evaluations. These metrics are standard and expected in similar work, and the reported results look promising. However, I still believe that at least one additional baseline beyond Safe-RLHF is crucial. I respectfully disagree with the claim that Safe-RLHF already demonstrates superior performance compared to [2] and [3]. [2] proposes stepwise DPO, which does not require either a reward model or a cost model and uses only two DPO runs, making it much more efficient and simpler than both Safe-RLHF and the proposed method. [3] also achieves higher helpfulness under similar harmfulness constraints. Furthermore, there are several other recent methods that have been shown to outperform Safe-RLHF. Comparing against such state-of-the-art baselines, in addition to Safe-RLHF, would substantially strengthen the experimental section.
> >
> > Overall, I appreciate the theoretical contribution articulated in Theorem 1. However, I remain cautious about the claims regarding the advantage of the proposed cost model and the strength of the experimental evidence.

---

> ### Author Response · Authors · 2025-11-27
> **Rebuttal Phase 2 Part 1**
>
> We sincerely thank the reviewer for their thoughtful feedback and continued evaluation of our paper.
>
> > *Question 1: I strongly recommend that the authors highlight all changes in the revised manuscript using a different color (e.g., blue), so that it is easier for reviewers to locate the modifications. In the current version, it is difficult to distinguish newly added results from the original content.*
>
>   We thank the reviewer for the helpful suggestion regarding readability. In the revised manuscript, all newly added results and modifications have been clearly highlighted in blue to ensure that the changes are easy to identify.
>
> > *Question 2 Part 1: Thank you for the additional experiments regarding the cost models. I agree that these results are important for demonstrating that the proposed cost model may outperform Safe-RLHF, and they should be moved to the main paper rather than remaining in Appendix D. Note that an extra page is available for the main paper during the rebuttal phase.*
>
> We appreciate the reviewer’s suggestion. We agree that these results strengthen the contribution and should be included in the main paper. At this stage, we have not uploaded the updated manuscript because we are still referencing the current numbering of results and tables in our responses to other reviewers. Uploading a partially updated version could shift this numbering and potentially cause confusion. In the final version, we will move the results from Appendix D to the main paper.
>
> > *Question 2 Part 2: Despite these supporting results, I still have several concerns. First, based on the confusion matrix of the Safe-RLHF cost model, the reported accuracy of Safe-RLHF appears to be only 57.8%. However, in the original Safe-RLHF paper (Table 1), the reported safety classification accuracy is around 85%, which is substantially higher than the value reported by the authors. Since both results are obtained using their training and test splits, this raises a concern about whether the comparison is truly fair. I would appreciate a clear explanation of this discrepancy.*
>
> We understand the confusion and the doubt is very valid, however, Table 1 in Safe RLHF  is not applicable here.  Specifically, we have not evaluated the responses rather the cost models only. We will explain how we sampled 1k examples which will clarify the apparent confusion.  Both the models (Safe RLHF-cost model, and our cost model) were evaluated on exactly the same set of test samples (prompts and response). The safe-RLHF test split consists of 5 important columns, the prompt, response 1, response 2, safety label for response 1 and safety label for response 2 (Preference based dataset). To construct this evaluation set, we sampled 1,000 examples randomly from the Safe-RLHF test split. Each prompt in this dataset contains two labeled responses out of which we sample one with equal probability. In this way we create 1000 samples consisting of 3 tuples (prompt, sampled response, sampled response’s safety-label (ground truth)). Our sampling procedure introduced randomness at two points: (i) we randomly selected 1,000 prompts from the full test set, and (ii) for each selected prompt, we randomly chose one of its two annotated responses while preserving the original ground-truth safety-label.  After constructing this set of 1,000 prompt–response pairs, we provided identical inputs to both the Safe-RLHF cost model and our CS-RLHF cost model. Each model assigned a cost score to each example, and these scores were evaluated against the ground-truth labels.
>
> Our result shows that our cost model is effective in classifying. In particular, the safe RLHF cost model has a higher false positive rate (classifying unsafe even if it is safe).
>
> In terms of responses, we have  table 17, 18, and19 (for response quality check) and table 14, 16,and 24 (for cost model scoring accuracy check) which shows that our model performs better. Due to the time constraints, we could not compare the responses from the CS-RLHF and the safe RLHF on the specific safe RLHF test split. We will do that in the final version.
>
> > *Question 2 Part 4: Finally, there is a minor typo at line 294: “we solve equation equation 10”; one occurrence of “equation” should be removed.
>
> We appreciate the reviewer for pointing out this typo, and we have corrected it.

---

> > ### Author Response · Authors · 2025-11-27
> > **Rebuttal Phase 2 Part 2**
> >
> > > *Question 2 Part 3: Moreover, I remain unconvinced by the claim that the proposed cost model is superior to Safe-RLHF. Examining Equation (10), it seems quite similar to the second term of Equation (4) in the original Safe-RLHF paper, which corresponds to the loss used to train the cost model to correctly predict safety labels. The proposed method appears to correspond to the case where we do not have access to pairwise preference information over responses, whereas Safe-RLHF assumes such preferences are available....*
> >
> > Empirically, we observe that our cost model outperforms the cost model of Safe RLHF (table 14, 16, and 24).
> >
> > Coming to the second point, we respectfully disagree with the reviewer on the second term of Safe-RLHF being equivalent to our equation (10). The second term of equation (4) from Safe-RLHF is ,
> >
> > $-E_{(x,y_{w},y_{l},s_{w},s_{l}) \sim D_{C}}\left[\log \sigma(s_{w}.C_{\psi}(x,y_{w}) + \log(\sigma(s_{l}.C_{\psi}(x,y_{l}))\right]$.
> >
> > Our equation (10) is:
> > $L_{CSRLHF}(\xi)= -E\left[t\log(F_{\xi}(x, y)) + (1-t)\log (1-F_{\xi}(x, y)))\right]$ (Although we have used $C_{\psi}$ in the paper, here we denote our cost model with $F_{\xi}$ and Safe-RLHF model with $C_{\psi}$ to demonstrate that the models are not similar neither in terms of parameterization nor in terms of output response, hence same symbol and parameters might create confusion.)
> >
> > How are we different?
> >
> > (i) We do not have preference based responses. So, a question arises is how to incorporate $y_{w}$ and $y_{l}$?
> >
> > A perfectly acceptable option(which seems to be the reviewer’s suggestion as well) is to make $y_{w} =y_{l} = y$. However, if we do so, then by Safe-RLHF nomenclature, both $s_{w} = s_{l} =1 ~or -1$ (Both would have same label). And so in that case, the second term of eq. (4) in Safe-RLHF would become $-E_{(x,y_{w},y_{l},s_{w},s_{l}) \sim D_{C})}\left[2 \log(\sigma(C_{\psi}(x,y)))\right]$ which is not equivalent to our equation (10).
> >
> > (ii) Irrelevance against misclassification although having $s_{w}+s_{l} =1$ (always) and $s_{g} = 1$ when harmful and $s_{g}=0$ where g ={w,l} when helpful:
> >
> > Let us consider there is a misclassification by our model thus, $t = 1$ and $F_{\xi}(x,y) = 0$, then by $L_{CSRLHF}(\xi)= -E\left[t\log(F_{\xi}(x, y)) + (1-t)\log (1-F_{\xi}(x, y))\right]$, we observe that the function blows-up.
> > However, with
> >  $L_{CE}(\psi)= -E\left[\log \sigma(s_{w}.C_{\psi}(x, y)) + \log \sigma(s_{l}.C_{\psi}(x, y))\right]$, $s_{w} =1$ and $C_{\psi}(x,y) = 0$,$\sigma(s_{w}.C_{\psi}(x, y))=0.5$ and $\log(0.5)$ is a finite constant value.  $L_{CE}(\psi) = 2 \log (0.5)$ and not blow up.
> >
> > Where is the difference?
> >
> > Note for our model the sigmoid layer is attached to the scorer head, thus our output value $F_{\xi}$ is different from$C_{\psi}$.
> >
> > Will attaching a sigmoid head to Safe-RLHF model solve the case?
> >
> > The answer is still no, as evident from our explanation above. Also because $L_{CE}(\psi)= -E\left[\log \sigma(s_{w}.C_{\psi}(x, y)) + \log \sigma(s_{l}.C_{\psi}(x, y))\right]$ is not equivalent to $L_{CE}(\psi)= -E\left[s_{w}.\log \sigma(C_{\psi}(x, y)) + s_{l}.\log \sigma(C_{\psi}(x, y))\right]$.
> >
> > In summary, the two functions are not the same.
> >
> > >*Question 3: I appreciate the authors providing the labeling protocol. However, I would like more clarification on inter-annotator agreement, annotation guidelines...... Please refer to...similar information to their paper.*
> >
> > We thank the reviewer for the thoughtful feedback. All prompt–response pairs in our dataset were reviewed by human annotators on our team and assigned a binary safety label based on semantic meaning—safe for responses whose intent is benign despite containing sensitive keywords, and unsafe for responses that encourage or endorse harmful behavior. To ensure consistency, all annotations were cross-validated by multiple annotators, and disagreements were resolved through consensus. We also conducted basic inter-annotator agreement, meaning that multiple annotators labeled overlapping subsets of examples and we verified that the assigned labels were consistent before resolving any disagreements. Importantly, our dataset is intentionally enriched with jailbreak-style prompts.
> >
> > While Safe-RLHF Appendix E.5 describes a large-scale, industrial annotation pipeline, our annotation setup was intentionally more lightweight because the goal of this work was to construct a targeted dataset suitable for evaluating our model rather than to replicate Safe-RLHF’s scale. This streamlined process allowed us to focus on collecting semantically meaningful safety violations without introducing unnecessary complexity. We envision that in future work, our model can be tested and scaled to much larger datasets as part of a broader evaluation framework.
> > Importantly, empirical evaluation confirms that this simple but targeted dataset provided the performance we aimed for. See results( for response see table: 17, 18, and19 and for cost model scoring check table 14and 24 ).

---

> > > ### Author Response · Authors · 2025-11-27
> > > **Rebuttal Phase 2 Part 3**
> > >
> > > >*Question 4: Thank you for providing the dataset statistics. Since the dataset is constructed from a subset of prompts from Safe-RLHF, the authors should explicitly acknowledge this in the paper..... In addition, this size is quite limited compared to Safe-RLHF, and this limitation should be clearly discussed.*
> > >
> > > We thank the reviewer for this helpful suggestion. We have now explicitly acknowledged in the paper that a subset of examples from the Safe-RLHF dataset was used. The Safe-RLHF dataset is released under the Apache 2.0 license, and we have verified that our usage is fully compliant with its terms, including proper attribution. The dataset size previously reported was 1,500 examples; we have since expanded it to 2,500 examples, and this updated size is now clearly stated in the main paper. We have also added a discussion of the dataset’s relatively limited scale compared to Safe-RLHF in the limitations section, along with our plan to further expand the dataset in future work.
> > >
> > > >*Question 5: I appreciate the experiments on a larger test set and the inclusion of win-rate/Elo evaluations. These metrics are standard and expected in similar work, and the reported results look promising. However, I still believe that at least one additional baseline beyond Safe-RLHF is crucial. ...Furthermore, there are several other recent methods that have been shown to outperform Safe-RLHF. Comparing against such state-of-the-art baselines, in addition to Safe-RLHF, would substantially strengthen the experimental section.*
> > >
> > > According to the reviewer’s suggestion, we have experimented with the baseline beyond safe-rlhf.  We compare the CS-RLHF with Stepwise DPO known as SACPO (Stepwise Alignment for Constrained Language Model Policy Optimization). We have given the identical set of prompts to both policy models and generated the responses. After the responses were generated, we used the Reward model (common to both Safe-RLHF and CS-RLHF) to generate reward scores and then convert the reward scores in to the sigmoid reward scores ($r_{i}^{dpo}$ for SACPO responses and $r_{i}^{cs}$ for CS-RLHF responses) and CS-RLHF’s cost model to score the responses generated from both models ($c_{i}^{dpo}$ for SACPO responses and $c_{i}^{cs}$ for CS-RLHF responses).
> > >
> > > We then create a combined cost score metric:
> > > $S_{i}^{g}=r_{i}^{g} - \lambda*\max((c_{i}^{g}-0.5),0)$
> > >
> > > We chose $\lambda=20$ which was also our chosen value for policy training, and 0.5 as the baseline for the cost classification and $g= \(dpo,cs\)$ respectively. This score combinedly portrays the safety and helpfulness of the model. The more the value of $S_{i}^{g}$ the better the performance is. The range of $S_{i}^{g} \in \[-10,1\]$ is bounded.
> > >
> > > Out of 1172 (same prompts used in ELO and win-rate test), SACPO reports better score ($S_{k}^{dpo}$) in 271 cases whereas CS-RLHF attains better performance in terms of combined score ($S_{k}^{cs}$) in 901 cases.
> > >
> > > To further understand this part and to check by how much each model is beating the other, we create a cumulative regret plot (cumulative-regret for t cases ($R_{i=1}^{t}$)).
> > >
> > > $R_{i=1}^{K} =  \sum_{i=0}^{K} (S^{*} - S^{g}_{i})$
> > >
> > > Where $S^{*}$ denotes the best score possible = 1 (maximum sigmoid reward and cost score in safety threshold, so 0).
> > >
> > > Thus, after finding the cumulative regret, we create a plot (Appendix M, Figure 6) which clearly shows that we outperform SACPO substantially by a considerable scale. From the regret plot we observed that, the cumulative regret for SACPO is **20.4%** higher than CS-RLHF. (Lower the regret better the model performance).
> > >
> > > In terms of safety comparisons SACPO reports 922 safe responses out of 1172 and CS-RLHF reports 919 safe responses out of 1172 responses which shows both the algorithms have comparable ability to produce safe responses.
> > > Finally, out of the 922 safe responses in SACPO, 783 were helpful and out of 1172 (including both safe and unsafe), 930 were helpful for SACPO. On the other hand for CS-RLHF out of 919  safe responses, 827 were helpful and out of 1172, 987 were helpful responses for CS-RLHF. Based on the helpfulness rates, the corresponding average sigmoid reward scores are approximately 0.43 for SACPO and 0.62 for CS-RLHF. Taken together, these results indicate that although both models produce a similar number of safe responses, CS-RLHF generates responses that are substantially more helpful overall, achieving a better balance between safety and helpfulness.

---

> > > > ### Comment · Reviewer_1fxE · 2025-11-27
> > > > **Official Comment by Reviewer 1fxE**
> > > >
> > > > I thank the authors for their reply. However, I still have several concerns.
> > > >
> > > > 1. I appreciate the effort to revise the manuscript and provide additional experimental results. I understand that ongoing changes can create confusion in the numbering. However, by the end of the discussion period we should be able to review a fully updated version in order to assess whether the paper is ready for publication. Although I am very positive about Theorem 1, I do not believe the paper is ready in its current form, and we currently have no way to verify that all necessary changes will be made after the rebuttal phase.
> > > >
> > > > 2. Thank you for detailing how the evaluation set is constructed. However, the procedure you describe, sampling prompts and then sampling responses, seems quite standard. My understanding is that Safe-RLHF uses a similar way to evaluate the accuracy of its cost model. Such randomness should not degrade performance as dramatically as reported (from 85% down to 57%). If it does, I am concerned that the 1,000 samples may inadvertently be cherry-picked.
> > > >
> > > >     To explain why I strongly doubt this result: I am very familiar with the Safe-RLHF dataset, including training their reward/cost model. I have evaluated their cost model multiple times using the full test set and both responses, rather than random subsampling, i.e., over all (prompt, response, safety label) tuples. Based on this experience, an accuracy of only 57% seems implausibly low, and I believe Safe-RLHF’s Table 1 is computed in this full-data manner. While I acknowledge that my own experience may be biased, Reviewer k7uw has raised a similar concern. I would be very interested in hearing Reviewer k7uw’s perspective and remain open to further discussion on this point.
> > > >
> > > > 3. I appreciate the mathematical comparison between the Safe-RLHF formulation and the proposed cost model. I agree that Equation (10) are not mathematically equivalent to the Safe-RLHF objective. However, I want to emphasize that both terms are designed to predict the safety label. They may differ in exact form and numerical values, but the intended functionality is the same. Since the authors’ results show that the proposed cost model substantially outperforms Safe-RLHF, I would like to understand the **theoretical** reason behind this improvement, beyond the empirical evidence. Why does your formulation provide stronger guarantees than Safe-RLHF? Under standard RLHF assumptions, the Safe-RLHF loss is consistent with the Bradley–Terry (BT) model, which is a common modeling assumption for reward/cost learning. What is the theoretical advantage of the proposed objective relative to this BT-based formulation?
> > > >
> > > > Overall, I remain very positive about Theorem 1, but I am not yet convinced by the cost-model. In addition, I do not think the paper, in its current form, is ready for publication. I strongly encourage the authors to provide a fully revised version before the discussion deadline so that we can evaluate the submission fairly.

---

> > > > > ### Author Response · Authors · 2025-11-28
> > > > > **Part 1 response of Rebuttal Phase 3**
> > > > >
> > > > > We thank the reviewer for continued engagement.
> > > > >
> > > > > >*In addition, I do not think the paper, in its current form, is ready for publication. I strongly encourage the authors to provide a fully revised version before the discussion deadline so that we can evaluate the submission fairly.*
> > > > >
> > > > > Just to reiterate, we have incorporated all requested updates into the manuscript; these changes are currently reflected in the appendix. We have not yet moved the material in Appendix D into the main paper, as one of the reviewers has not had a chance to read our response. Once that reviewer has responded, we will promote the corresponding content from Appendix D into the main paper as appropriate.
> > > > >
> > > > > Below, we summarize the modifications made. All new or revised content is highlighted in blue in the PDF. **We kindly ask the reviewer to review these changes and confirm that they adequately address the concerns.**
> > > > >
> > > > > **Changes in the Main Paper**
> > > > >
> > > > > Page 2 (Introduction): We now explicitly state the dataset size.
> > > > >
> > > > > Page 10 (Acknowledgments): We acknowledge that we use a subset of the Safe-RLHF dataset.
> > > > >
> > > > > **Changes in the Supplementary Material**
> > > > >
> > > > > Page 3, Appendix A (Limitations and Future Work):
> > > > > We explicitly note that our dataset is smaller than the Safe-RLHF dataset, identify this as a limitation of the current work, and state our plan to expand the dataset in future iterations.
> > > > >
> > > > > Pages 16–17, Appendix C (XS-Test and Win-Rate Study):
> > > > >
> > > > > Table 4: Safe-RLHF responses
> > > > >
> > > > > Table 5: CS-RLHF responses
> > > > >
> > > > > Table 6: Win rate and Elo results
> > > > >
> > > > > Pages 18–19, Appendix D (Cost Model Comparison on 1000 Safe-RLHF Test Split):
> > > > >
> > > > > Table 7: Safe-RLHF cost model
> > > > >
> > > > > Table 8: CS-RLHF cost model
> > > > >
> > > > > Page 32:
> > > > > New prompt–response example added as Table 17 (caption in blue).
> > > > >
> > > > > Page 33, Appendix M (Rebuttal Phase 2 Experimental Results on Stepwise DPO and CS-RLHF):
> > > > >
> > > > > Text additions in blue
> > > > >
> > > > > Figure 6 (caption in blue)
> > > > >
> > > > > Page 40:
> > > > > Table 23: CS-RLHF failure cases (caption in blue).
> > > > >
> > > > > Page 41:
> > > > > Table 24: XS-Test benchmark examples where identical inputs are given to both cost models to compare their scores (entire table in blue).
> > > > >
> > > > > >*Regarding the result on the Safe RLHF test split dataset.*
> > > > >
> > > > > * Our results are correct, and the test set was not cherry-picked.*  The apparent discrepancy may be due to different choices of hyperparameters, in particular, the threshold used to classify a response as harmful or not. We now clarify our evaluation protocol in detail.
> > > > >
> > > > > * When evaluating the cost models, we use a threshold of
> > > > > 0.5 (i.e., $\sigma(0)$). We first sample 1,000 examples uniformly at random from the Safe-RLHF test split. Each prompt in this dataset comes with two labeled responses; for each prompt, we randomly select one of the two with equal probability. This yields a set of 1,000 prompt–response pairs. We then feed these identical pairs to both the Safe-RLHF cost model and our CS-RLHF cost model, obtain a cost score for each example, and classify a response as safe if its score is below
> > > > > 0.5 and unsafe otherwise. We report the resulting metrics based on this fixed threshold, which we selected after careful deliberation. *All of these details are now documented in the paper so that the results can be reproduced.*
> > > > >
> > > > > * As the reviewer has extensive experience with the Safe-RLHF dataset, we would be happy to adopt any alternative, reasonable protocol for constructing the test set that they may suggest and report the corresponding results. We also note that, not only on the Safe-RLHF test split but also on our own constructed dataset, *Safe RLHF consistently underperforms CS-RLHF.*

---

> > > > > > ### Author Response · Authors · 2025-11-28
> > > > > > **Rebuttal Phase 3 Part 2**
> > > > > >
> > > > > > >*Regarding the improved performance*
> > > > > >
> > > > > > * The reviewer’s comment here is based on a misunderstanding. *We do use a Bradley–Terry (BT) model, but only for the reward model, not for the cost model.* This design choice is deliberate. Our objective is to solve the optimization problem in (2), where the constraint only requires that the expected harmfulness score remain below a given threshold. *Thus, we do not need to minimize the harmfulness score itself;* we only need to ensure that it stays below the safety threshold. Consequently, a BT-style preference model for costs is unnecessary—we only require a model that can reliably determine whether a response’s cost is below or above the threshold. This insight motivates our cost-model design, which does not rely on a preference dataset for costs and instead uses harmful/harmless labels.
> > > > > >
> > > > > > *In essence, our approach differs from Safe RLHF in three key ways:*
> > > > > >
> > > > > > **Cost modeling:** Our cost model does not require pairwise preference data; harmful/harmless labels are sufficient, since we only need to enforce the safety constraint. We have also enriched the dataset with Jailbreaking prompts and empirically found that our cost model is less sensitive to the keywords.
> > > > > >
> > > > > > **Training-time guarantees:** Primal–dual methods, as used in Safe RLHF, cannot guarantee a precise balance between safety and helpfulness, whereas our ReLU-based formulation provides such a guarantee, as established in Theorem 1.
> > > > > >
> > > > > > **Inference-time guarantees:** We further provide inference-time performance and safety guarantees via the BoN-based selection mechanism, as formalized in Corollary 1.
> > > > > >
> > > > > > Overall, both our cost model and our training approach are fundamentally different from those in Safe RLHF. We provide explicit guarantees that carefully balance optimality (helpfulness) and constraint satisfaction, as formalized in our theory. Our empirical results further demonstrate that, compared to Safe RLHF, our approach achieves better performance under the same safety constraints. All corresponding clarifications and additions have been incorporated into the manuscript and are highlighted in blue. *In light of this, we urge the reviewer to kindly reconsider their position.*

---

> > > > > > > ### Comment · Reviewer_1fxE · 2025-11-28
> > > > > > > **Official Comment by Reviewer 1fxE**
> > > > > > >
> > > > > > > I thank the authors for their reply.
> > > > > > >
> > > > > > > 1. I understand that the authors have incorporated the updates into the appendix. However, integrating some of the key results into the main paper (e.g., the cost model results, ELO analysis, and SACPO results) would require substantial revisions, especially to the experimental section. In this sense, my claim that the paper, in its current form, is not yet ready for publication remains valid.
> > > > > > >
> > > > > > > 2. I understand the full procedure used to construct the test set, so there is no need to repeat it again. If I remember correctly, the Safe RLHF cost model classifies a response as safe when its score is **below 0** and unsafe otherwise, rather than using a threshold of 0.5. I believe this mismatch in the threshold is likely the main reason for the low accuracy reported for Safe RLHF. Please use 0 as the threshold for the Safe RLHF cost model.
> > > > > > >
> > > > > > > 3. I do not misunderstand your method. I mentioned the Bradley–Terry (B–T) model only because Safe RLHF builds on it, and I view this as a theoretical advantage of their formulation: it aligns with the standard RLHF assumptions on human preferences. I recognize that your approach is fundamentally different from Safe RLHF and does not require preference data. However, being different does not automatically make it better, and I do not see “not requiring preferences” as a strong advantage on its own, especially since, as I noted earlier, Safe RLHF can in principle rely on its second term alone when preference data are unavailable.
> > > > > > > What I would like to understand more clearly is: beyond the empirical results, what are the **theoretical** benefits of the proposed **cost model**? The training-time and inference-time guarantees you present do not appear to follow directly from the specific cost model itself.
> > > > > > >
> > > > > > > I do not want to repeat these points again and again. I believe I am taking a fair stance: I am willing to raise my score if the authors adequately address my concerns about the cost model. I will also closely follow the other reviewers’ feedback. If they conclude that my concerns are unfounded or have been properly resolved, I will update my score accordingly.

---

### Official Review · Reviewer_k7uw · 2025-10-29

**Soundness:** 3
**Presentation:** 3
**Contribution:** 3
**Rating:** 4
**Confidence:** 3

**Summary:**

The paper tackles LLM safety as a constrained optimization problem and argues that standard CMDP/Lagrangian RLHF is fragile: performance hinges on noisy reward/cost scoring and expensive dual-variable tuning that lacks fixed-λ guarantees and can be jailbroken. It proposes Certifiable Safe-RLHF (CS-RLHF), which (i) trains a cost model on large harmful/harmless labels to deliver semantically grounded safety scores, and (ii) replaces the Lagrangian with a rectified exact-penalty objective that activates only when the expected cost exceeds a threshold. (iii) The authors also add a BoN decode-time guarantee.

**Strengths:**

The formalization of safety alignment as a constrained optimization problem is highly novel, making the safety alignment process itself more reliable and trustworthy.

Highlighting that an overly conservative cost model can negatively impact the final performance of aligned LLMs is crucial. While the issue of overly conservative models has been well-studied in the RL community, it remains underexplored in the LLM space. This paper makes an important contribution by addressing this gap.

Additionally, the introduction of the new Best-of-N (BoN) sampler provides a decode-time safety guarantee, further enhancing the trustworthiness of the safety alignment process.

**Weaknesses:**

Please refer to the Questions.

**Questions:**

1) **Expectation vs. pointwise constraint (Eq. 4).**
   The rectified penalty appears to be applied to an *expected* cost. How exactly is ReLU composed with the expectation—\(\mathrm{ReLU}(\mathbb{E}[C]-d)\) vs. \(\mathbb{E}[\mathrm{ReLU}(C-d)]\)? These are not equivalent; please clarify the chosen form and its implications for per-sample violations.

2) **Penalty scaling and “non-negotiable” safety (Thm. 1 & line 233).**
   The guarantee seems to require \(\lambda\) to increase as the target \(\epsilon\) decreases, potentially affecting the reward–cost trade-off. How is \(\lambda\) set in practice? If feasibility loosens for moderate \(\lambda\), how does this align with the claim that “harmlessness is non-negotiable”?

3) **Dataset comparability and fairness.**
   How does your dataset differ from Safe-RLHF’s? Since Safe-RLHF also provides binary safe/unsafe labels, why not train/evaluate on their data for comparisons? Using different datasets risks confounding the empirical gains.

4) **Behavior on jailbreaks (Fig. 2).**
   Figure 2 seems to show the cost model assigning low cost to certain jailbreak prompts (classifying them as safe). Please explain this behavior, provide failure cases, and quantify its frequency.

5) **Cost-model design vs. simpler alternatives.**
   You fine-tune LLaMA-2-7B-chat as the cost model. A simpler baseline—e.g., using layer-26 activations with an unsupervised classifier to output \(\{0,1\}\)—could test whether heavy fine-tuning is necessary. If such a baseline performs similarly, Eq. (10) may be unnecessarily complex.

6) **Loss in Eq. (10).**
   Why not use a standard classification loss (e.g., logistic cross-entropy or margin losses)? Please justify the chosen objective or show ablations against standard losses.

7) **Semantics over keywords: evaluation gaps (line 309).**
   To support the claim that the cost model captures semantics rather than triggers, evaluatation on over-refusal sets such as *XSTest*[1], *CoCoNot*[2], and *OR-Bench*[3], where benign prompts share toxic tokens with unsafe prompts. This would more directly test the stated research question.

[1] Röttger, Paul, et al. "Xstest: A test suite for identifying exaggerated safety behaviours in large language models." arXiv preprint arXiv:2308.01263 (2023).

[2] Brahman, Faeze, et al. "The art of saying no: Contextual noncompliance in language models." Advances in Neural Information Processing Systems 37 (2024): 49706-49748.

[3] Cui, Justin, et al. "Or-bench: An over-refusal benchmark for large language models." arXiv preprint arXiv:2405.20947 (2024).


I would like to raise my score if part of the questions are answered appropriately.

---

> ### Author Response · Authors · 2025-11-21
> **Rebuttal part 1**
>
> We appreciate your time and thoughtful evaluation of our paper. We recap your comment and present our detailed response as follows. We would also be happy to provide further clarifications if suitable.
>
> >*Expectation vs. pointwise constraint (Eq. 4)*
>
> Our training procedure does not provide formal guarantees at the per-sample level; it only enforces safety in expectation. Enforcing safety on a per-sample basis during training is substantially more challenging than enforcing it in expectation: it amounts to a worst-case, pointwise constraint over an enormous response space, leading to highly non-smooth and brittle optimization (effectively optimizing the maximum violation). Note that Safe RLHF does not even provide guarantees in expectation. Moreover, although we do not establish per-sample guarantees during training, we can enforce safety guarantees at inference time using the BoN approach as described in the paper.
>
> >*Penalty scaling and “non-negotiable” safety (Thm. 1 & line 233).*
>
> The reviewer is correct to point out that theoretically $\lambda$ should increase as $\epsilon$ decreases. However, this is also true for any other constrained problem where one needs to tune the hyperparameter as the requirement changes. Note that compared to the traditional primal-dual-based approach, the major strength is that we do not need to tune the $\lambda$ variable even for a single value of $\epsilon$.  We have also **provided ablation studies (in Appendix B)** on how **different values of $\lambda$ affect our performance**. We find that **a reasonable value of $\lambda$ is enough to ensure safety**. The primary objective of Theorem 1 is to demonstrate the closeness of the policy returned by our method to the optimal policy. We know that the primal-dual algorithm converges to the sub-optimal policy. However, since we are not solving the primal-dual problem here, we need to demonstrate that our optimized policy is close to the optimal one of the original problem. **For our empirical study, we do not tune the lambda throughout the learning process**.
>
> >* How does your dataset differ from Safe-RLHF’s?*
>
> Our **dataset** is more **concentrated with jailbreaking prompts**. To demonstrate the benefits of having a rectified cost function, we **inculcate a large number of carefully crafted jailbreak prompts**. Further, **we eliminate the need for alternative responses, $y_{w}$ and $y_{l}$**. Hence, each **data point** is a **prompt, response pair, and the response class (safe/unsafe)**
>
> >*Since Safe-RLHF also provides binary safe/unsafe labels, why not train/evaluate on their data for comparisons?*
>
> We agree with the reviewer that the data in Safe-RLHF is also binary. We have also included a subset of the dataset in our mix. **However, we have performed additional experiments**, using the **training set of Safe-RLHF** and **testing 1,000 data points from the Safe-RLHF test set prompts**. The **confusion matrix for Safe-RLHF cost model verdict and CS-RLHF cost model verdict are as given below**
> | (**Ground truth ↓**) & (**Safe-RLHF verdict →**) | **Predicted safe** | **Predicted unsafe** |
> |-----------------------------|--------------------|----------------------|
> | **Actually safe**           | 44/466                 | 422/466                  |
> | **Actually unsafe**         | 0/534                | 534/534                 |
>
> The Safe-RLHF cost model's confusion matrix over **1000** test samples is shown above
>
> | (**Ground truth ↓**) & (**CS-RLHF verdict →**)| **Predicted safe** | **Predicted unsafe** |
> |-----------------------|--------------------|----------------------|
> | **Actually safe**  | 444/466                 | 22/466                  |
> | **Actually unsafe**| 8/534                | 526/534                 |
>
> The CS-RLHF cost model's confusion matrix over **1000** test samples is shown above.
>  For more details, refer to **Appendix D**
>
> >* Behavior on jailbreaks (Fig. 2).* (part 1)
>
> The reviewer might be mistaken. Specifically, we evaluate the responses to the prompts, rather than the prompts themselves. The model assigns a low cost to responses from our model for jailbreaking prompts and classifies them as safe. This happens because, although the prompt itself is adversarial (i.e., it attempts to elicit harmful information), our CS-RLHF policy refuses to comply and instead produces a safe, non-harmful response.
> **We would like to draw the reviewer's attention to a specific prompt in Table 17, prompt 1**. _Here, the user attempts a jailbreak by posing a hypothetical scenario to extract sensitive information. CS-RLHF detects the intent and responds with a fictitious, harmless answer—helpful yet safe. In contrast, Safe-RLHF provides a more realistic response. Our goal is to enable LLMs to deliver valid, useful answers without risking sensitive information disclosure_.
>
> We found multiple such instances. Hence, the low cost score.

---

> ### Author Response · Authors · 2025-11-21
> **Rebuttal part 2**
>
> > provide failure cases: (continuation of Question 4 concluding parts)
>
> We have added another table which summarises the failure cases as per the reviewer's suggestion. **Please check table 23**
>
> >quantify its frequency:
>
> _CS-RLHF_
>
> **For jailbreaking prompts** failure occurs **14/120** cases **Figure 2**.
>
> **For regular prompts** failure occurs **9/500** cases **Figure 2**
>
> _Safe-RLHF_
>
> **For jailbreaking prompts** failure occurs **98/120** cases **Figure 2**.
>
> **For regular prompts** failure occurs **65/500** cases **Figure 2**
>
> _ALPACA-7B_
>
> **For jailbreaking prompts** failure occurs **115/120** cases **Figure 2**.
>
> **For regular prompts** failure occurs **255/500** cases **Figure 2**
>
>   >*Cost-model design vs. simpler alternatives.*
>
> We **implemented a simpler baseline** that **utilizes only the layer-26 hidden activations of LLaMA-2-7B-chat** as thereviewer suggested, together with a **lightweight classifier head**, while keeping the entire model frozen. This serves as a **non-fine-tuned** alternative (say **Model A**) to **our full CS-RLHF cost model (say Model B)**. When evaluated on the same held-out validation split, Model B achieves 98% accuracy, whereas Model A reaches only 52% accuracy, effectively performing at near-chance levels. This suggests relying solely on internal activations is insufficient for intent-aware safety classification. The observation of the two models' performance on **120 (60 safe and 60 unsafe) randomly selected prompts** from our mix produces the following results (Due to time constraints, we could not test on higher numbers. However, if the reviewer feels satisfied, we will perform a detailed study in the final version)
>
> |Ground truth$\downarrow$|Model A classification|Model B classification|
> |--------------------|-----------------------|--------------------|
> |Safe|60/60|60/60|
> |Unsafe|0/60|60/60|
>
> In other words,**Model A** predicts all inputs as safe, which is not correct.
> and
>
> >*Loss in Eq. (10).*
>
> Equation (10) indeed simplifies to the standard cross-entropy loss applied to a sigmoid output, where the model takes a prompt–response pair and its label as input during training. After training, the model outputs a single scalar cost score for each such pair. **We have simplified it now**.
>
> >*XSTest, CoCoNot, and OR-Bench*
>
> We **evaluated both cost models** on **XSTest dataset**, which contains **250 benign prompts** that include toxic-looking keywords and **200 genuinely harmful prompts**.
>
> $XSTest Score = \alpha*\frac{correct-responses}{total-safe-prompts} + (1-\alpha)*\frac{correct-refusal}{unsafe-prompts}$
>
> |**Cost model$\downarrow$**|**XS score**|**correctly answered and non-refused**|**correct refusals**|
> |---------------------------|-------------------------|------------------------------|-----------------------|
> |CS-RLHF|0.9105|225|187|
> |Safe-RLHF|0.0714|18|14|
> |Human evaluation of the prompts|0.8873|221|179|
>
> Table A: Lists the XS-test Score on responses generated from Safe-RLHF optimized policy model scored by the above cost inference models
>
> |**Cost model$\downarrow$**|**XS score**|**correctly answered and non-refused**|**correct refusals**|
> |---------------------------|-------------------------|------------------------------|-----------------------|
> |CS-RLHF|0.9643|241|193|
> |Safe-RLHF|0.0714|86|51|
> |Human evaluation of the prompts|0.8873|225|191|
>
> Table B: Lists the XS-test Score on responses generated from the CS-RLHF optimized policy model scored by the above cost inference models
>
> Explanation: CS-RLHF cost model achieves scores of 0.91–0.96, closely matching human agreement and correctly identifying 225–241 of the 250 benign prompts and 187–193 of the 200 unsafe prompts. In contrast, the Safe-RLHF cost model collapses on this benchmark (scores 0.07–0.32), misclassifying the majority of benign prompts due to keyword triggers. These results demonstrate that the CS-RLHF cost model captures semantic intent rather than surface-level keywords, directly addressing the reviewer’s request. For further details check appendix C.
>
> Because of the time constraint, we could not evaluate on OR-Bench and CoCoNot. We will do that in the final version.

---

> > ### Comment · Reviewer_k7uw · 2025-11-26
> >
> > Thank you for the authors’ detailed response and for conducting additional experiments. The new results and clarifications help strengthen several aspects of the submission.
> > That said, I remain unconvinced about the theoretical justification and the design of the cost model. My concerns are aligned with the second point raised in the Official Comment by Reviewer 1fxE, which I believe accurately captures the issues I still find insufficiently addressed. Additionally, regarding Safe RLHF, the authors state that their formulation is lack of expectation-based guarantees; however, in the arXiv version of Safe RLHF, Equation (11) includes the expectation operator, suggesting that Safe RLHF satisfies expectation-based constraints.
> > If the authors can properly address these points—particularly the theoretical validity of their assumptions and the precise distinction between their cost model and expectation-based guarantees—I would be willing to raise my score.

---

> > > ### Author Response · Authors · 2025-11-27
> > > **Rebuttal Phase 2 Part 1**
> > >
> > > >*Question1 Part1: That said, I remain unconvinced about the theoretical justification and the design of the cost model. My concerns are aligned with the second point raised in the Official Comment by Reviewer 1fxE, which I believe accurately captures the issues I still find insufficiently addressed.*
> > >
> > > We understand the confusion and the doubt is very valid, however, Table 1 in Safe RLHF  is not applicable here.  Specifically, we have not evaluated the responses rather the cost models only. We will explain how we sampled 1k examples which will clarify the apparent confusion.  Both the models (Safe RLHF-cost model, and our cost model) were evaluated on exactly the same set of test samples (prompts and response). The safe-RLHF test split consists of 5 important columns, the prompt, response 1, response 2, safety label for response 1 and safety label for response 2 (Preference based dataset). To construct this evaluation set, we sampled 1,000 examples randomly from the Safe-RLHF test split. Each prompt in this dataset contains two labeled responses out of which we sample one with equal probability. In this way we create 1000 samples consisting of 3 tuples (prompt, sampled response, sampled response’s safety-label (ground truth)). Our sampling procedure introduced randomness at two points: (i) we randomly selected 1,000 prompts from the full test set, and (ii) for each selected prompt, we randomly chose one of its two annotated responses while preserving the original ground-truth safety-label.  After constructing this set of 1,000 prompt–response pairs, we provided identical inputs to both the Safe-RLHF cost model and our CS-RLHF cost model. Each model assigned a cost score to each example, and these scores were evaluated against the ground-truth labels.
> > >
> > >
> > > Our result shows that our cost model is effective in classifying. In particular, the safe RLHF cost model has a higher false positive rate (classifying unsafe even if it is safe).
> > >
> > > In terms of responses, we have  table 17, 18, and19 (for response quality check) and table 14, 16,and 24 (for cost model scoring accuracy check) which shows that our model performs better. Due to the time constraints, we could not compare the responses from the CS-RLHF and the safe RLHF on the specific safe RLHF test split. We will do that in the final version.

---

> > > > ### Author Response · Authors · 2025-11-27
> > > > **Rebuttal Phase 2 Part 2**
> > > >
> > > > >*Question1 Part2:*
> > > >
> > > >
> > > > Empirically, we observe that our cost model outperforms the cost model of Safe RLHF (table 14, 16, and 24).
> > > >
> > > >
> > > > Coming to the second point, we respectfully disagree with the reviewer on the second term of Safe-RLHF being equivalent to our equation (10). The second term of equation (4) from Safe-RLHF is ,
> > > >
> > > >
> > > > $-E_{(x,y_{w},y_{l},s_{w},s_{l}) \sim D_{C}}\left[\log \sigma(s_{w}.C_{\psi}(x,y_{w}) + \log(\sigma(s_{l}.C_{\psi}(x,y_{l}))\right]$.
> > > >
> > > >
> > > > Our equation (10) is:
> > > > $L_{CSRLHF}(\xi)= -E\left[t\log(F_{\xi}(x, y)) + (1-t)\log (1-F_{\xi}(x, y)))\right]$ (Although we have used $C_{\psi}$ in the paper, here we denote our cost model with $F_{\xi}$ and Safe-RLHF model with $C_{\psi}$ to demonstrate that the models are not similar neither in terms of parameterization nor in terms of output response, hence same symbol and parameters might create confusion.)
> > > >
> > > >
> > > > How are we different?
> > > >
> > > >
> > > > (i) We do not have preference based responses. So, a question arises is how to incorporate $y_{w}$ and $y_{l}$?
> > > >
> > > >
> > > > A perfectly acceptable option(which seems to be the reviewer’s suggestion as well) is to make $y_{w} =y_{l} = y$. However, if we do so, then by Safe-RLHF nomenclature, both $s_{w} = s_{l} =1 ~or -1$ (Both would have same label). And so in that case, the second term of eq. (4) in Safe-RLHF would become $-E_{(x,y_{w},y_{l},s_{w},s_{l}) \sim D_{C})}\left[2 \log(\sigma(C_{\psi}(x,y)))\right]$ which is not equivalent to our equation (10).
> > > >
> > > >
> > > > (ii) Irrelevance against misclassification although having $s_{w}+s_{l} =1$ (always) and $s_{g} = 1$ when harmful and $s_{g}=0$ where g ={w,l} when helpful:
> > > >
> > > >
> > > > Let us consider there is a misclassification by our model thus, $t = 1$ and $F_{\xi}(x,y) = 0$, then by $L_{CSRLHF}(\xi)= -E\left[t\log(F_{\xi}(x, y)) + (1-t)\log (1-F_{\xi}(x, y))\right]$, we observe that the function blows-up.
> > > > However, with
> > > >  $L_{CE}(\psi)= -E\left[\log \sigma(s_{w}.C_{\psi}(x, y)) + \log \sigma(s_{l}.C_{\psi}(x, y))\right]$, $s_{w} =1$ and $C_{\psi}(x,y) = 0$,$\sigma(s_{w}.C_{\psi}(x, y))=0.5$ and $\log(0.5)$ is a finite constant value.  $L_{CE}(\psi) = 2 \log (0.5)$ and not blow up.
> > > >
> > > >
> > > > Where is the difference?
> > > >
> > > >
> > > > Note for our model the sigmoid layer is attached to the scorer head, thus our output value $F_{\xi}$ is different from$C_{\psi}$.
> > > >
> > > >
> > > > Will attaching a sigmoid head to Safe-RLHF model solve the case?
> > > >
> > > >
> > > > The answer is still no, as evident from our explanation above. Also because $L_{CE}(\psi)= -E\left[\log \sigma(s_{w}.C_{\psi}(x, y)) + \log \sigma(s_{l}.C_{\psi}(x, y))\right]$ is not equivalent to $L_{CE}(\psi)= -E\left[s_{w}.\log \sigma(C_{\psi}(x, y)) + s_{l}.\log \sigma(C_{\psi}(x, y))\right]$.
> > > >
> > > >
> > > > In summary, the two functions are not the same.

---

> > > > > ### Author Response · Authors · 2025-11-27
> > > > > **Rebuttal Phase 2 Part 3**
> > > > >
> > > > > >*Question2: Additionally, regarding Safe RLHF, the authors state that their formulation is lack of expectation-based guarantees; however, in the arXiv version of Safe RLHF, Equation (11) includes the expectation operator, suggesting that Safe RLHF satisfies expectation-based constraints. If the authors can properly address these points—particularly the theoretical validity of their assumptions and the precise distinction between their cost model and expectation-based guarantees*
> > > > >
> > > > >  Safe RLHF indeed imposes an expectation-based constraint, as we do. However, their proposed method does not provide any theoretical guarantee of constraint satisfaction (even in expectation). We elaborate on this point. In the Safe RLHF paper, the primal–dual scheme increases the dual variable whenever the constraint is violated at a given training iteration and decreases it otherwise. In the CMDP literature, such primal–dual methods only guarantee constraint satisfaction and performance bounds for the averaged primal iterates [A1]. For large-scale models such as LLMs, averaging parameters over iterations is computationally expensive and practically infeasible, and thus Safe RLHF, as instantiated for LLMs, does not offer a formal constraint or performance guarantee.
> > > > >
> > > > > Moreover, while one could set a very large dual value to heuristically enforce the constraint, this typically leads to highly conservative behavior and a severely suboptimal helpfulness score. Conversely, using a small dual value may fail to satisfy the constraint. In contrast, our ReLU-based penalty formulation has the key advantage that, once a response is safe, its penalty term becomes zero; increasing the dual value then only further penalizes unsafe responses without degrading the score of safe, helpful ones. As shown in Theorem 1, this allows us to achieve constraint satisfaction guarantees without sacrificing helpfulness.
> > > > >
> > > > > Furthermore, we provide inference time safety guarantees using the BoN approach (and using softBoN when the model is imperfect) which safe RLHF does not provide.
> > > > >
> > > > > [A1]. Moskovitz, T., O’Donoghue, B., Veeriah, V., Flennerhag, S., Singh, S. and Zahavy, T., 2023, July. Reload: Reinforcement learning with optimistic ascent-descent for last-iterate convergence in constrained mdps. In International Conference on Machine Learning (pp. 25303-25336). PMLR.

---

### Official Review · Reviewer_JxHU · 2025-10-30

**Soundness:** 2
**Presentation:** 2
**Contribution:** 2
**Rating:** 2
**Confidence:** 4

**Summary:**

Their method modifies SafeRLHF by using a fixed penalty term instead of the lagrangian to encourage safety. They also use ReLU to only take positive penalties into account instead of negative ones. They show that using their penalty during best of N inference guarantees that the safe response is selected if any of the responses are safe. They also create a dataset of labeled safe and unsafe prompt response pairs.

**Strengths:**

Novelty is sufficient but mixed.
Their core contribution is a small change to an existing method. However, I think the other contributions add up to sufficient novelty.

The experimental results are solid and show improvement over a reasonable baseline.

**Weaknesses:**

The guarantee that “if any of the N candidates is safe, the selected response is safe” seems trivial to achieve. The following method does the trivially: Pick the safe response which gets the highest reward. Let me know if I’m missing something, since I didn’t understand this part very well (see below).

Equation 10 isn’t well motivated. The paper doesn’t explain the difference between equation 10, and training the classifier using SFT with sigmoid cross entropy, or the advantage of their approach over sigmoid cross entropy.


Lack of ablation studies

There aren’t any ablation studies so it’s hard to tell which of the contributions improved the results over the baseline. I would like to see ablation studies testing the impact of each contribution individually. Examples of possible ablation studies:
Using their classifier with the baseline optimization method.
Training a classifier on their dataset with sigmoid cross entropy instead of their objective function.
Comparing training a classifier on their dataset of safe and unsafe responses with training on an existing dataset.


Clarity

The inference time safety section isn’t well explained. Specifically: How is equation 11 analogous to BoN? Note: The explanation of the BoN guarantee in the introduction is much more clear.

Tables and figures are hard to understand.
Figure 1 is hard to read because the text is too small, and it’s hard to tell what is a different method and what’s a different part of the same method.
Table 2 confusing. It’s hard to tell what subset of the data each of the numbers is computed over. It would be more straightforward to have one table for evaluating the safety of each method's responses according to humans, and a separate table for how often each of the classifiers agree with humans. For example, “Reward Evaluation Helpful” is a subset of “Human Verdict Safe” but that’s not immediately obvious without doing math, or reading the caption in detail.
The text in figure 2 is too small to read. Also, the description for each of the subplots should be included in the figure itself, not just in the caption for ease of reading. Each subplot should use the same color for ‘regular prompts’ so the legend can be shared between all of them.

In the Cost Model section, it’s not explained why the baseline heavily penalizes keywords.

Minor: ReLU is defined multiple times in 4.1

**Questions:**

See weaknesses. Specifically, my questions around BoN, and equation 10.

---

> ### Author Response · Authors · 2025-11-21
> **Rebuttal Part I**
>
> We appreciate your time and thoughtful evaluation of our paper. We recap your comment and present our detailed response as follows. We would also be happy to provide further clarifications if suitable.
>
> >*Equation 10 isn’t well motivated. ..cross entropy.*
>
> We agree with the reviewer, Equation (10) indeed simplifies to the standard cross-entropy loss applied to a sigmoid output, where the model takes a prompt–response pair and its label as input during training. After training, the model outputs a single scalar cost score for each such pair. **We have changed it in the current version now.**
>
> >*Lack of ablation studies There aren’t any ablation … dataset.*
>
> We would respectfully point out to the reviewer that we provided an ablation study in the supplementary section, Appendix B, which aims to show the effect of selecting different $\lambda$ values. We would also like to highlight that the study mentions other comparisons as well, such as the comparison between the cost models with their optimization method and our objective policy optimization methods in table 2, some explicit prompt response pair and their respective scores in **Table 14** under appendix (L.1). We have **now** conducted a detailed study on the XStest score prompts [1] on our proposed cost model and the cost model proposed in Safe RLHF. The details of which can be found in appendix C.
>
> >*Training a classifier on their dataset with sigmoid cross entropy instead of their objective function.*
>
> We performed this experiment as per the reviewer's suggestion and trained our cost model over Safe-RLHF training set, further we tested the outcome over 1000 prompts from the Safe-RLHF test set. The confusion matrix for Safe-RLHF cost model verdict and CS-RLHF cost model verdict are as given below
> | (**Ground truth↓**) & (**Safe-RLHF verdict →**) | **Predicted safe** | **Predicted unsafe** |
> |-----------------------------|--------------------|----------------------|
> | **Actually safe**           | 44/466                 | 422/466                  |
> | **Actually unsafe**         | 0/534                | 534/534                 |
>
> The Safe-RLHF cost model's confusion matrix over **1000** test samples is shown above
>
> | (**Ground truth ↓**) & (**CS-RLHF verdict →**)| **Predicted safe** | **Predicted unsafe** |
> |-----------------------|--------------------|----------------------|
> | **Actually safe**  | 444/466                 | 22/466                  |
> | **Actually unsafe**| 8/534                | 526/534                 |
>
> The CS-RLHF cost model's confusion matrix over **1000** test samples is shown above. It shows that our cost model is better compared to the safe RLHF.
>  For more details, refer to Appendix D
>
> >*...How is equation 11 analogous to BoN? Note: The explanation of the BoN guarantee in the introduction is much more clear.*
>
> Equation (11) is not analogous to BoN. We use (11) specifically to select actions at inference time. Concretely, given a prompt x, we generate N candidate responses $y_i$ ​ (as in BoN), and then select the response according to $\arg\max_{y_i} r(x,y_i)-\lambda ReLu[c(x,y_i)]$.
>
> In Corollary 1, we provide the corresponding feasibility guarantee.
>
> The reviewer’s intuition is almost correct: ideally, we would like to pick the highest-reward response among those that are safe. However, it may happen that none of the sampled responses are certified safe under the learned cost model, which is why we instead use the penalized objective above as discussed in equation (11). Moreover, the reward and cost models themselves can be imperfect. We will address this in the final version.
>
> To address potential model errors, we further consider a softmax relaxation of the max operator and show that, even when the reward and cost models are misspecified, this softmax-based selector still yields a response whose performance is close to that of the optimal one.
>
> [1] Paul Röttger, Hannah Rose Kirk, Bertie Vidgen, Giuseppe Attanasio, Federico Bianchi, and Dirk Hovy. Xstest: A test suite for identifying exaggerated safety behaviours in large language models, 2024. URL https://arxiv.org/abs/2308.01263.

---

> ### Author Response · Authors · 2025-11-21
> **Rebuttal part 2**
>
> >*Tables and figures are hard to understand...all of them.*
>
> We thank the reviewer for pointing out the readability challenges. *We have revised these elements according to the reviewer's suggestion.*
>
> *A brief explanation of the elements are as follows*
>
> 1) **Figure 1 provides an overview of our full CS-RLHF training pipeline**, including **supervised fine-tuning, data collection, reward and cost model construction, and the final RLHF optimization stage**. _The figure illustrates how user prompts and model responses flow through each component—first generating preference data, then training the reward and cost models, and finally using both signals to optimize the policy_.
>
> 2) **Table 2** presents a controlled **comparison of CS-RLHF and the baseline across 100 test prompts**, evaluating both the accuracy of the two cost models and the actual safety and helpfulness of the responses generated by each method. (The tree representation is as below).
>
> CS-RLHF vs Safe-RLHF (100 evaluation prompts)
>
> │
>
> ├── Safe-RLHF responses evaluated by CS-RLHF cost model
>
> │   ├── 76 / 100 predicted safe
>
> │   │   ├── 73 confirmed safe (human)
>
> │   │   └── 3 unsafe (human)
>
> │   └── Baseline cost model (Safe-RLHF cost model) self-evaluation: 11 / 100 safe (all correct)
>
> │
>
> ├── CS-RLHF responses evaluated by CS-RLHF cost model
>
> │   ├── 92 / 100 predicted safe
>
> │   │   ├── 90 confirmed safe (human)
>
> │   │   └── 2 unsafe (human)
>
> │   └── Baseline cost model (Safe-RLHF cost model) evaluation: 29 / 100 safe
>
> │       ├── 26 confirmed safe
>
> │       └── 3 unsafe
>
> │
>
> └── Helpfulness comparison (only among human-confirmed safe responses)
>
>     ├── CS-RLHF: 87 helpful, 3 unhelpful
>
>     └── Safe-RLHF: 69 helpful, 4 unhelpful
>
>
> The evaluation across 100 prompts shows that 76/100 Safe-RLHF policy-generated responses and 92/100 CS-RLHF policy-generated responses were predicted as safe by the cost model, with human judges confirming 73/76 and 90/92 as genuinely safe, respectively. Of those 73 true positive Safe-RLHF responses and 90 true positive CS-RLHF responses, the helpfulness ratings were 69/73 and 87/90, respectively.
>
> 3) In response to the reviewer’s readability feedback, **we revised Figure 2** so that all regular-prompt responses appear in a single consistent color across all subplots, enabling a shared legend. We also increased text size, clarified visual elements, and added explicit subplot titles within each panel. These changes significantly improve readability and make the differences between methods more visually apparent.
> Figure 2 shows the cost–reward trade-offs for Alpaca-7B, the baseline Safe-RLHF model, and CS-RLHF across 500 regular and 120 jailbreak prompts. **The black dot corresponds to Jailbreak prompt's response scores, and the blue dot represents normal prompt's response scores. The green region is the safe zone**. Ideally, most responses should lie in the top left corner, that is the second quadrant.
>
> >*In the Cost Model section, it’s not explained why the baseline heavily penalizes keywords.*
>
> Our **empirical analysis suggests** that the **baseline cost model** heavily **penalizes keywords (Appendix L, table 14)**. _Our intuition is that this behaviour is due to the presence of training data such that both pairs are unsafe_; however, the model is trained to prefer the less harmful one. This behavior has been observed in many data points. Thus, there is a chance that the model's learning is impacted. It is also observed that harmful responses frequently **contain sensitive tokens** (e.g., “bypass,” “lockpick,” “poison,” “exploit”); these tokens **appear proportionately in the “non-preferred’’  and "preferred" responses**. Thus, that might be a reason for such behavior.
>
> **One major observation that we want to bring to the reviewer's notice is the improvement this pair of optimizations (our cost model and policy optimization technique) makes**. For example, **observe the responses generated for the 1st jailbreaking prompt in Table 17**. _Observe here, the user tries to jailbreak by creating a hypothetical situation and trying to get sensitive information through trickery. CS-RLHF senses it and, for a fictitious question, produces a fictitious answer. So, in a way, it is helpful directly, but also the answer produced is safe, knowing that real-world people do not have trouble with this suggestion, which is a fair suggestion. So in a way, it is helpful but not harmful. In contrast, the answer produced by Safe-RLHF is more realistic. This is what we aim to do in this paper: to make the LLM intelligent enough to give a valid response without the risk of passing sensitive information_.
>
> >*Minor: ReLU is defined multiple times in 4.1*
>
> We thank the reviewer for pointing this out. We have made the changes.

---

### Author Response · Authors · 2025-12-03
**Summary of our contributions, our discussion, and the difference of our approach with the Safe-RLHF: Part1**

In this generic response, we are going to point out our contributions, the changes we have made incorporating the reviewers' comments, and point out the main distinction from our work and Safe RLHF work as multiple reviewers have asked for.

**Our Contributions:**

This work makes several substantial contributions to safe reinforcement learning from human feedback for large language models. **First**, we introduce a meticulously curated safety dataset collected from diverse sources, **including GPT-4o, DeepSeek, other open-source LLMs, a subset of Safe-RLHF training prompts, and additional prompts inspired by recent academic literature** on prompt perturbations, persona-conditioned language models, and toxicity analyses. **Our dataset construction pipeline applies semantic perturbations to seed prompts and aggregates multiple responses from different models**, yielding a rich distribution of safe, borderline, and unsafe behaviors (details in Appendix I). **Second**, through systematic experiments (Tables 1, 15, 16), we identify a **central limitation of the Safe-RLHF cost model—its heavy reliance on keyword-triggered scoring** rather than genuine understanding of the underlying intent. We propose a simpler, more effective, and more easily trainable alternative that **aligns significantly better with human judgment**. This is demonstrated across comprehensive empirical evaluations, including pairwise comparisons, **human-supervised scoring, XS-Test, and win-rate/Elo analyses (Tables 2, 3(XS-Test), 5(Elo), (table for response quality check (18, 19, 20)(Table for cost scoring(15, 16, 25);** Figures 2–3.

**Third**, we develop a penalty-based optimization framework for the **constrained objective**:
$\max_{\theta} J_R(\theta) - \lambda\mathrm{ReLU}\big(J_C(\theta)\big)$,
which strongly penalizes safety violations. **We provide theoretical analysis showing that this procedure converges to an **$ \epsilon$**-suboptimal policy under the safety constraint in Theorem 1**, and we validate its effectiveness empirically across a wide range of benchmarks (Tables 2–5, 15–25; Figures 2–4). **Finally**, while our training procedure enforces safety in expectation—a formulation that is far more tractable than per-sample worst-case constraints—**we additionally provide inference-time safety guarantees using a Best-of-N sampling strategy**. This allows us to enforce high-probability per-sample safety at deployment time, offering practical guarantees that Safe-RLHF does not provide even in expectation.


**All the reviewers have appreciated our contributions.** For example, reviewer JxHU mentioned that *`` I think the other contributions add up to sufficient novelty. The experimental results are solid and show improvement over a reasonable baseline."*

Reviewer k7uw mentioned that *``The formalization of safety alignment as a constrained optimization problem is highly novel, making the safety alignment process itself more reliable and trustworthy."*


Reviewer 1fxE mentioned that *``Under the formulation in Eq. 5 and Theorem 1, the optimal policy enjoys a higher expected return than the Lagrangian baseline, and the dual parameter is bounded and tied to constraint violations. This is the paper’s primary contribution. Moving from the Lagrangian setup to the proposed framework is well motivated and theoretically characterized."*

The reviewer has also raised some concerns. In the following, we summarize how we addressed those.

>*Addressing Reviewers' concerns.*

**Reviewer JxHU** requested clarification of the inference-time BoN guarantee, the formulation of Eq. (10), and specific ablations. We clarified that Eq. (10) is standard sigmoid cross-entropy and updated the manuscript accordingly . Moreover, we added all requested ablations, **including $\lambda\$-sensitivity (Appendix B) and additional comparisons (Appendix L.1)**, and we conducted a **full semantic-robustness evaluation on XS-Test (Table 3)**, where CS-RLHF substantially outperforms Safe-RLHF. Reviewer JxHU  also mentioned an ablation, which we addressed by evaluating our classifier on the Safe-RLHF train/test split, where it **achieves significantly higher correctness, especially on safe examples (Appendix D, Tables 8–9)**. We further clarified that Eq. (11) is not BoN, but a penalized inference-time selector that **offers a formal safety guarantee via Corollary 1**, and **we introduced a soft-BoN variant for robustness** (Explanation: https://openreview.net/forum?id=LwdxjLUa64&noteId=v8UGKhOpBa). Finally, consistent with Reviewer’s request, we demonstrated that Safe-RLHF’s BT term induces keyword-triggered failures, whereas CS-RLHF remains contextually grounded, as shown in our intuition-driven analysis and empirical results (Appendix L, Tables 15, 16, 18, 25). (Explanation: https://openreview.net/forum?id=LwdxjLUa64&noteId=gJyLBJGg8R).

---

> ### Author Response · Authors · 2025-12-03
> **Part 2: Addressing the Reviewers' concerns (continued)**
>
> **Reviewer k7uw** focused on $\lambda\$, dataset fairness, fine-tuning necessity, and semantic generalization. We clarified that $\lambda\$ does not require tuning, since **Theorem 1 guarantees correctness for any sufficiently large $\lambda\$**, and $\lambda\$-ablations are reported in Appendix B and extended in Appendix L.1 (Explanation: https://openreview.net/forum?id=LwdxjLUa64&noteId=YNHd89120p). For fairness, Reviewer requested a comparison on the Safe-RLHF dataset, which is already addressed above and in (Appendix D, Table 8-9). Reviewer  also questioned the need for fine-tuning; **we added a no-fine-tuning baseline (52% accuracy), showing that meaningful safety behavior cannot be captured** by shallow probes, whereas **our fine-tuned model achieves 98%**. Finally, they requested semantic-safety benchmark results; **our XS-Test evaluation shows CS-RLHF scoring 0.91–0.96, while Safe-RLHF collapses to 0.07–0.32 (Table 3)**. (Explanation : https://openreview.net/forum?id=LwdxjLUa64&noteId=KZjjfJGysa). **Reviewer k7uw’s Phase-2 comments overlapped with Reviewer 1fxE’s**, and we clarified that the ~85% result reported in Safe-RLHF (Table 1) is not comparable, as it measures paired preferences rather than binary correctness—under a matched 1,000-prompt binary evaluation, Safe-RLHF achieves ≈57.8%, whereas CS-RLHF performs far better (Appendix D; Tables 8–9, 15–16, 25; Tables 18–20) (Explanation: https://openreview.net/forum?id=LwdxjLUa64&noteId=f7iiz9FVIx). **We also showed that Safe-RLHF’s BT+classification loss is mathematically distinct from ours** (Explanation: https://openreview.net/forum?id=LwdxjLUa64&noteId=IorpqwFVQn), **and that its primal–dual training lacks any final-iterate guarantee, unlike our rectified-penalty formulation** (Explanation: https://openreview.net/forum?id=LwdxjLUa64&noteId=U94ZNcc2xo).
>
> **Reviewer 1fxE** asked for a matched evaluation on the Safe-RLHF split, clearer dataset details, **scaling to 1,000 prompts, inclusion of SACPO (Stepwise-DPO), and Elo/win-rate evaluations**. All of these requests were fully addressed in our rebuttal. The 1,000-prompt Safe-RLHF evaluation with confusion matrices is provided in Appendix D, Tables 8–9 (Explanation: https://openreview.net/forum?id=LwdxjLUa64&noteId=mzM2Qdv82U). We expanded dataset documentation in Appendix I, and **after scaling to 1,000 prompts, CS-RLHF is both safer and more helpful** (Table 2; Explanation: https://openreview.net/forum?id=LwdxjLUa64&noteId=pBxPXWhoXP). For the stronger baseline, **CS-RLHF outperforms SACPO 901 vs. 271, achieving 20.4% lower cumulative regret** (Appendix M, Fig. 6; Explanation: https://openreview.net/forum?id=LwdxjLUa64&noteId=jU5LEIq5Bh). Reviewer  also requested **interpretability metrics; CS-RLHF achieves a 0.544 helpfulness win-rate (Elo +30.7) and a 0.595 harmlessness win-rate (Elo +66.6) over Safe-RLHF (Table 5)**. Phase-2 concerns were identical to Reviewer k7uw’s and were fully addressed, with all **new additions highlighted in blue** (Explanation: https://openreview.net/forum?id=LwdxjLUa64&noteId=CzH4RBWCrB). (Explanation: https://openreview.net/forum?id=LwdxjLUa64&noteId=xwGSHH0xjG). In addition, to accommodate the reviewer1fxE’s request and remain within the one-page limit, we merged the CS-RLHF vs. SACPO (Stepwise-DPO) win-rate and Elo **results into a single consolidated table in the main paper**. As shown in Table 5, CS-RLHF achieves higher helpfulness (0.5998, Elo +70.28) and harmlessness (0.5192, Elo +13.33) than SACPO, providing a clear and concise comparison.

---

> ### Author Response · Authors · 2025-12-03
> **Part 3: Distinction between Safe RLHF and the CS-RLHF**
>
> Beyond individual reviewer points, **we emphasized the core conceptual distinctions between CS-RLHF and Safe-RLHF**: (i) **CS-RLHF removes the Bradley–Terry preference term,** eliminating Safe-RLHF’s structural misclassification of safe responses; (ii) **our rectified-penalty formulation provides a formal safety guarantee (Theorem 1)**, which Safe-RLHF’s primal–dual updates cannot ensure at LLM scale; and (iii) CS-RLHF uniquely **offers inference-time safety via BoN and soft-BoN (Corollary 1)**—a guarantee absent in Safe-RLHF (Explanation: https://openreview.net/forum?id=LwdxjLUa64&noteId=q6M6rrSOEM).
>
> **Importantly** the reviewers (especially, reviewer 1fxE) have some doubts regarding the fundamental difference between the Safe RLHF cost model and the CS-RLHF cost model. We could not continue the discussion due to the ban of the author-reviewer discussion. We want to elaborate more on this. *A critical difference between the Safe RLHF and CS-RLH is that* Safe-RLHF cost model is trained on a preference dataset over harmful and harmless responses. In particular, in addition to a binary label indicating whether each response is harmful, the dataset for Safe RLHF also includes pairwise annotations specifying which of two responses is more harmful. Safe RLHF combines a Bradley–Terry (BT)–style preference term with a classification term. This introduces a potential issue. Consider a pair where both responses are safe, but y_1​ is annotated as more harmful than y_2. The classification component encourages both responses to have low (safe) cost scores, whereas the BT component pushes the score of y_1 ​ upward relative to y_2​, potentially to a level that crosses the “unsafe” threshold despite y_1​ being safe.
>
> As a result, a moderately safe response can be assigned a high cost and misclassified as unsafe; we have observed such behavior empirically in our dataset. To avoid this structural issue, we adopt a simplified cost model that focuses solely on classifying responses as harmful or harmless, without enforcing pairwise ordering within the safe cluster. This sacrifices some granularity among safe responses but makes the sign of the cost more robust: safe responses are much less likely to be assigned high (unsafe) cost values, which is more appropriate for downstream safety constraints. Importantly, our goal is not to finely rank safe responses but to obtain a cost signal whose sign reliably distinguishes safe from unsafe content for use in risk-constrained optimization. (See Result in Table 25)
> Finally, combining the cost model with our novel rectified penalty-based approach (with provable theoretical properties) helps us to achieve a better overall safety.
>
>
> Overall, we believe that (we could not verify as the discussion was terminated due to the ICLR's revised policy) we have addressed all the main concerns of the reviewers and have also updated the paper (with the changes marked in Blue). We earnestly request the AC to decide based on our overall contributions and additional results and discussions we provided.

---

### Meta-Review · Area_Chair_S8jA · 2026-01-05

**Summary:**

This paper proposes improvements to the Safe RLHF framework, including a new cost model and a rectified penalty-based optimization method. Although the authors conducted supplementary experiments during the rebuttal phase, the core issues raised by the reviewers remain unresolved.

1) The proposed cost model can be viewed as a special case of Safe-RLHF when pairwise preference data is unavailable, lacking substantial theoretical justification for its advantage.

2) In the key performance comparison experiment, an incorrect classification threshold (0.5 instead of the original 0) was applied to the Safe-RLHF cost model, rendering the reported performance superiority invalid.

3) Critical experimental results remain scattered in the appendix, the main text lacks a complete argument, and a final evaluable version was not formed within the review cycle.

Due to methodological errors in the core experimental comparison, lack of sufficient theoretical innovation, and incomplete manuscript structure, the decision is to reject the submission.

**Reviewer Concerns:**

Addressed Reviewer Concerns：

The authors effectively addressed several technical concerns through supplementary work: They expanded the experimental evaluation with large-scale testing, win-rate and Elo analysis, and introduced SACPO as a comparative baseline. Theoretically, they clarified the distinction between expectation-based training constraints and inference-time BoN guarantees, while refining the derivation of the rectified penalty term. Additionally, they improved figure readability and supplemented dataset statistics and model implementation details.

Unresolved Critical Issues：

The paper contains the following fatal flaws: The core experiment used an incorrect threshold to evaluate the Safe-RLHF cost model, invalidating the claimed performance superiority, and the authors did not re-validate this during the review period. The proposed cost model was questioned as merely a special case of the existing framework when preference data is missing, and its theoretical novelty failed to be convincingly demonstrated. The manuscript structure remains incomplete, with key experimental results scattered in the appendix and not integrated into a coherent, prioritized argument.

**Reviewer Scores:**

JxHU (Initial Score: 2)
Projected Final Score: 2
This reviewer raised fundamental concerns about the clarity of the methodological foundation and the completeness of the experiments. Although the authors supplemented some analyses, the core experimental flaws and theoretical disputes exposed in subsequent discussions failed to address these underlying concerns.

k7uw (Initial Score: 4)
Projected Final Score: 3-4
While the authors addressed some technical issues, this reviewer explicitly aligned with another reviewer's criticism of the cost model comparison and theoretical justification in the later discussion phase. As fatal issues such as the threshold error in the core experiment remained uncorrected, the score would remain at or below the threshold.

1fxE (Initial Score: 4)
Projected Final Score: 3
This reviewer acknowledged the theoretical contribution but strongly questioned the novelty of the cost model. Although the authors supplemented extensive experiments, the reviewer ultimately highlighted two unresolved core issues: the invalid comparison due to incorrect thresholds and the lack of theoretical advantage in the cost model.


The discussion revealed fatal unresolved issues (invalid core experiment, weak theoretical innovation) that outweighed the incremental improvements in the rebuttal. Key reviewers' scores are projected to decline overall, strengthening the consensus for rejection.

---

### Decision · Program_Chairs · 2026-01-26

Reject